# CONFORMAL COORDINATE FRAMES FOR DISENTANGLEMENT

**Edmond Cunningham**
Department of Computer Science
University of Massachusetts Amherst
Amherst, MA 01002, USA
edmondcunnin@cs.umass.edu

## ABSTRACT

Disentangled representations are central to interpretable machine learning, yet learning them without supervision is unidentifiable. Conformal ICA, a special case of independent mechanism analysis (IMA), provides identifiability guarantees but is too restrictive to be practically useful. We propose to locally approximate conformal ICA by learning a conformal frame field that fits data, is integrable, and has implicit independent components. We enforce integrability and statistical independence through stochastic losses in a scalable way that require only Jacobian-vector products. On Neal's funnel distribution in dimensions 4 through 64, our approach consistently recovers the ground truth structure, demonstrating that local conformal frame fields offer a scalable foundation for geometrically grounded disentanglement.

## 1 INTRODUCTION

Learning disentangled representations is a central goal of representation learning (Bengio et al., 2013; Schölkopf et al., 2021), yet without supervision it is provably impossible without appropriate inductive biases (Locatello et al., 2020). Independent mechanism analysis (IMA) (Gresele et al., 2021; Matthes et al., 2025) provides such a bias by constraining the Jacobian of a generative map to factor as an orthogonal matrix times a diagonal matrix, avoiding many unidentifiability issues of general nonlinear ICA (Comon, 1994; Hyvärinen & Pajunen, 1999; Gresele et al., 2021; Buchholz et al., 2022). IMA has also been used to explain the disentanglement capabilities of certain VAEs (Reizinger et al., 2022). While it is not yet known whether IMA is fully identifiable, a special case where the generative map is conformal is identifiable up to trivial symmetries (Buchholz et al., 2022), meaning the generative model is essentially determined by the data distribution alone. However, Liouville's theorem restricts conformal maps in dimensions $n \geq 3$ to compositions of translations, rotations, dilations, and inversions (Hartman, 1958), so despite its identifiability guarantees, conformal ICA has seen little practical adoption.

Our key insight is that conformal structure can be enforced locally, so we abandon the global diffeomorphism and instead learn a local approximation where data is observed. We parameterize an orthogonal matrix $U(\mathbf{x}) \in O(n)$ and a conformal factor $\lambda(\mathbf{x}) > 0$ at each point $\mathbf{x}$, so that $J(\mathbf{x}) = \lambda(\mathbf{x})U(\mathbf{x})$ represents the Jacobian of a conformal map. Each column of $J(\mathbf{x})$ represents a disentangled direction (Matthes et al., 2025), and conformal identifiability leaves essentially no freedom in the choice of frame once the data density is fit. For $J(\mathbf{x})$ to be a valid Jacobian with implied independent components, we enforce integrability and statistical independence through scalable stochastic losses requiring only Jacobian-vector products, giving $O(n)$ complexity per sample. Although our approach cannot recover global latent variables, it recovers how latent variables act on observations, which suffices for data manipulation and has been proposed as a defining characteristic of disentanglement (Matthes et al., 2025). We evaluate on Neal's funnel distribution across dimensions 4 through 64 and find that our local approach consistently recovers the principal direction of variation where a global conformal baseline fails, suggesting that local conformal frame fields are a practical alternative to global conformal maps for disentanglement.

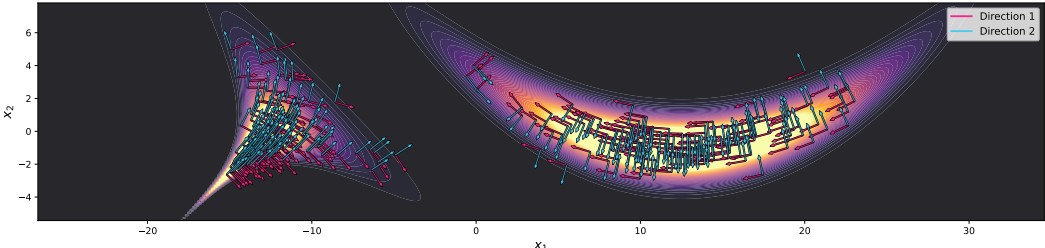

Figure 1: Learned conformal frame directions on a dataset sampled from a mixture of Neal's funnel and banana distributions. The arrows show the principal directions, which are the columns of $U(\mathbf{x})$. The frame directions are aligned with the structure of the data.

## 2   METHOD

A smooth, global frame is a tuple of vector fields that form a basis for the tangent space at each point in the data space (Lee, 2000). We represent a frame on the tangent bundle of Euclidean space as an invertible matrix $J(\mathbf{x}) \in GL(n)$ whose columns contain the standard components of the vector fields. A frame is *conformal* if it can be written as $J(\mathbf{x}) = \lambda(\mathbf{x})\,U(\mathbf{x})$, where $U(\mathbf{x}) \in O(n)$ and $\lambda(\mathbf{x}) > 0$. A frame is a *coordinate frame* if its basis vectors are partial derivatives of a diffeomorphism $f \colon \mathbf{z} \mapsto \mathbf{x}$, so that $J_i(\mathbf{x}) = \frac{\partial f}{\partial z_i}\big|_{\mathbf{z}=f^{-1}(\mathbf{x})}$. An arbitrary conformal frame need not be a coordinate frame, and even when it is, the implied latent variables $\mathbf{z}$ are not guaranteed to have independent components without further constraints. Our approach learns a conformal frame that is a coordinate frame with independent components, recovering the principal directions of variation in the data, as in Figure 1. In this section, we derive the constraints ensuring the learned conformal frame is a coordinate frame with independent components.

### 2.1   DISTRIBUTION MATCHING

To perform nonlinear ICA, the model must fit the observed data distribution. For a conformal map with Jacobian $J = \lambda U$, the change of variables formula relates the data density to the latent density through

$$\log p_{\mathbf{x}}(\mathbf{x}) = \log p_{\mathbf{z}}(f^{-1}(\mathbf{x})) - n \log \lambda(\mathbf{x}) \tag{1}$$

Since we have no global diffeomorphism through which to evaluate a nontrivial prior, we assume a uniform latent distribution, for which $\log p_{\mathbf{z}}$ is constant. Additionally, to avoid intractable normalizing constants, we fit our model to data by matching the score functions of the model and data distributions rather than the densities themselves.

$$\nabla_{\mathbf{x}} \log p_{\mathbf{x}}(\mathbf{x}) = -n \, \nabla_{\mathbf{x}} \log \lambda(\mathbf{x}) \tag{2}$$

We enforce equation 2 by minimizing the score matching loss $\mathcal{L}_{\text{score}} = \mathbb{E}_{\mathbf{x}}\big[\|\nabla_{\mathbf{x}} \log p_{\mathbf{x}}(\mathbf{x}) + n\nabla_{\mathbf{x}} \log \lambda(\mathbf{x})\|^2\big]$, where the data score $\nabla_{\mathbf{x}} \log p_{\mathbf{x}}$ is estimated via denoising score matching (Vincent, 2011).

The score matching objective does not involve $U(\mathbf{x})$ at all, so the score loss alone leaves the frame entirely unconstrained. The resolution comes from conformal identifiability (Buchholz et al., 2022), which guarantees that at most finitely many orthogonal frames yield a valid diffeomorphism with independent components once $\log \lambda(\mathbf{x})$ is fit to data. The integrability and independence losses introduced below select this essentially unique frame. We note that the identifiability result is proven for global conformal maps, and extending it to local frame fields remains open, but in practice our approach recovers consistent frame fields.

### 2.2   INTEGRABILITY

For the columns $J_i = \lambda U_i$ to be coordinate vector fields of a diffeomorphism $f$, meaning $\frac{\partial}{\partial z_i} = J_i^\top \nabla_{\mathbf{x}}$, the mixed partial derivatives must commute, i.e. $\partial J_i / \partial z_j = \partial J_j / \partial z_i$. Without access to latent coordinates, this condition is expressed as the vanishing of the Lie bracket (Lee, 2000),

$$[J_i, J_j](\mathbf{x}) = 0 \quad \text{for all } i \neq j \tag{3}$$

For numerical stability, we rewrite this condition in terms of the orthonormal frame and the gradient of $\log \lambda$ as follows:

**Proposition 1** (Conformal structure equation). *Let $J_i(\mathbf{x}) = \lambda(\mathbf{x}) \, U_i(\mathbf{x})$ where $U(\mathbf{x}) \in O(n)$ and $\lambda(\mathbf{x}) > 0$. Then $[J_i, J_j](\mathbf{x}) = 0$ for all $i \neq j$ if and only if*

$$[U_i, U_j](\mathbf{x}) = \left( U_j^\top(\mathbf{x}) \, \nabla_{\mathbf{x}} \log \lambda \right) U_i(\mathbf{x}) - \left( U_i^\top(\mathbf{x}) \, \nabla_{\mathbf{x}} \log \lambda \right) U_j(\mathbf{x}). \tag{4}$$

The proof is given in Appendix B. Let $V_{ij}$ denote the residual between the two sides of equation 4. A straightforward loss is

$$\mathcal{L}_{\text{int}} = \sum_{i<j} \| V_{ij}(\mathbf{x}) \|^2 \tag{5}$$

This loss sums over $\binom{n}{2}$ pairs at $O(n^3)$ cost, which we reduce to $O(n)$ by contracting against random probe vectors:

**Proposition 2** (Stochastic integrability loss). *Let $X, Y \sim \mathcal{N}(0, I_n)$ be independent and define $\hat{\mathcal{L}}_{int} = \frac{1}{2} \| \sum_{i,j} V_{ij} X_i Y_j \|^2$. Then $\hat{\mathcal{L}}_{int}$ is an unbiased estimator of $\mathcal{L}_{int}$ and satisfies $\hat{\mathcal{L}}_{int} = 0$ almost surely if and only if equation 4 holds for all pairs $i \neq j$. It can be evaluated using only Jacobian-vector products of $U$ and $\nabla_{\mathbf{x}} \log \lambda$, giving $O(n)$ complexity per sample.*

The proof and explicit form of $\hat{\mathcal{L}}_{\text{int}}$ are given in Appendix B.

### 2.3 INDEPENDENT COMPONENTS

The final constraint is that the implied latent variables $\mathbf{z}$ should have statistically independent components. From the change of variables formula equation 1 the implied latent log-density is $\phi(\mathbf{x}) = \log p_{\mathbf{x}}(\mathbf{x}) + n \log \lambda(\mathbf{x})$. For independent components, the Hessian of the latent log-density must be diagonal.

$$\frac{\partial^2 \phi}{\partial z_i \partial z_j}(\mathbf{x}) = 0 \quad \text{for all } i \neq j \tag{6}$$

where $\frac{\partial}{\partial z_i} = J_i^\top \nabla_{\mathbf{x}}$ is the latent derivative operator. Let $H_{ij} = \frac{\partial^2 \phi}{\partial z_i \partial z_j}$ be the Hessian of the latent log-density. A natural loss is $\mathcal{L}_{\text{ind}} = \sum_{i \neq j} H_{ij}^2$, but it requires $O(n^2)$ entries and the diagonal entries are generally nonzero. Let $\tilde{H}_{ij} = \lambda^{-2} J_i(J_j(\phi))$ be the scaled Hessian:

**Proposition 3** (Stochastic independence loss). *Let $X, Y \sim \mathcal{N}(0, I_n)$ and $M \sim Bernoulli(1/2)^n$ be independent, and form masked probes $X' = X \odot M$, $Y' = Y \odot (1 - M)$. The stochastic loss $\hat{\mathcal{L}}_{ind} = 4(X'^\top \tilde{H} Y')^2$ is an unbiased estimator of $\mathcal{L}_{ind}$ and can be evaluated in $O(n)$ time via Jacobian-vector products.*

The proof is given in Appendix B. Combining all three objectives, our full training loss is

$$\mathcal{L} = \mathcal{L}_{\text{score}} + \alpha_{\text{int}} \, \hat{\mathcal{L}}_{\text{int}} + \alpha_{\text{ind}} \, \hat{\mathcal{L}}_{\text{ind}} \tag{7}$$

where $\alpha_{\text{int}}$ and $\alpha_{\text{ind}}$ control the strength of the geometric constraints. All three terms can be estimated with $O(n)$ cost per sample using Jacobian-vector products, making the approach tractable in high dimensions.

## 3 EXPERIMENTS

In our experiments, we parameterize $U(\mathbf{x})$ with a neural network that outputs an unconstrained $n \times n$ matrix and obtain $U(\mathbf{x}) \in O(n)$ as the $Q$ factor of its QR decomposition, and we parameterize $\log \lambda(\mathbf{x})$ rather than $\lambda(\mathbf{x})$ directly for numerical stability. Our hypothesis is that the orthogonality and identifiability of conformal maps enable our approach to recover the principal direction of variation from data. To evaluate this, we train our model on a distribution with clear geometric structure and measure whether the learned frame recovers it. We use Neal's funnel distribution (Neal, 2003), which has a single axis of dominant variation whose scale controls the spread in all other dimensions. To prevent trivial solutions, we apply a random rotation so that the funnel axis is not aligned with any coordinate direction. We perform experiments in dimensions 4, 8, 16, 32, and 64 with 8 random seeds per dimension.

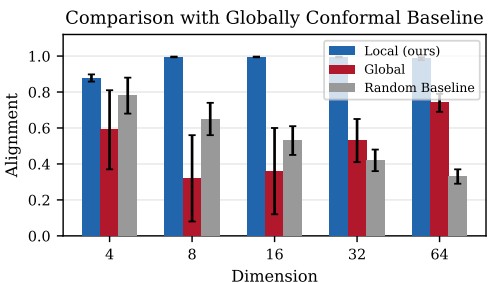
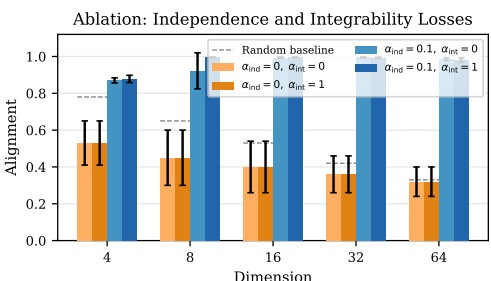

Figure 2: Alignment scores across dimensions for our local approach, a global conformal baseline, and a random baseline. Our method consistently recovers the funnel axis while the global baseline does not.

Figure 3: Ablation of the independent component and integrability loss weights. The independent component loss is critical for recovering the funnel axis, while the integrability loss seems less related to axis alignment.

We define the *alignment score* as the maximum absolute inner product between any learned frame direction and the true funnel axis, averaged over data points and seeds (1.0 is perfect recovery). As a reference, we include the alignment scores for a random frame as well.

We train our model with $\alpha_{ind} = 0.1$ and $\alpha_{int} = 1.0$. Our model consistently recovers the funnel axis across all tested dimensions with alignment above $0.87$, reaching $0.99$ in dimensions 16 and 32, significantly outperforming the random baseline, which falls steadily with dimension. We also compare against a global conformal baseline consisting of a single Möbius layer, which by Liouville's theorem is the most general globally conformal map in $n \geq 3$. As shown in Figure 2, the global baseline struggles to recover the principal direction, with alignment as low as $0.32$ in 8D and $0.36$ in 16D, falling below the random baseline. Our local conformal frame field outperforms the global baseline by $0.24$ to $0.63$ across all dimensions, confirming that global conformal maps are too rigid to capture the funnel geometry.

We ablate $\alpha_{ind} \in \{0, 0.1\}$ and $\alpha_{int} \in \{0, 1.0\}$ to isolate the contribution of each geometric constraint (Figure 3). The independence loss is critical for recovering the funnel axis. Without it ($\alpha_{ind} = 0$), alignment is near random regardless of $\alpha_{int}$. With $\alpha_{ind} = 0.1$, alignment is strong ($0.87$ to $0.99$) whether or not the integrability loss is active. Although the integrability loss does not substantially improve alignment, it is critical for ensuring that the learned frame does correspond to a valid coordinate system.

## 4  DISCUSSION AND FUTURE WORK

We have presented a local conformal frame approach to disentanglement that sidesteps the rigidity of global conformal maps, suggesting that the IMA principle can be made practical by working locally rather than globally. The most immediate limitation is that the independence loss requires access to the true log-density, which restricts the current method to settings where the likelihood is known. Extending the approach to the unsupervised regime will require a surrogate that estimates the relevant second-order latent structure from samples alone, but in the meantime the method can be applied to problems such as sampling from unnormalized densities. On the computational side, because our losses interact with $U(\mathbf{x})$ only through matrix-vector products, orthogonal convolutions (Singla & Feizi, 2021) could replace the dense QR parameterization and scale to high-dimensional structured inputs such as images. Looking ahead, we believe that local geometric constraints have the potential to enable scalable and identifiable unsupervised representation learning in complex, high-dimensional domains.

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

## A    RELATED WORK

**Conformal and orthogonal models.**   Several works have explored the role of orthogonality and conformal structure in generative models (Gropp et al., 2020; Horan et al., 2021). Ross & Cresswell (2021) considered the problem of learning conformal embeddings as a low-dimensional generative model of data. Our approach builds on this line of work by adopting the independent mechanism analysis (IMA) (Gresele et al., 2021) principle as an inductive bias, which constrains the Jacobian of the generative model to have orthogonal columns and connects to identifiability results (Buchholz et al., 2022). The class of normalizing flows that satisfy this principle are principal component flows (PCFs) (Cunningham et al., 2022), but this approach requires parameterizing an explicit global diffeomorphism using normalizing flows. This flow-based approach does not scale because the IMA principle requires the Jacobian to be orthogonal, which is not compatible with the typical Jacobian structure of normalizing flows. Our approach avoids these limitations by directly parameterizing a local frame field that satisfies conformal structure by construction, without requiring a global diffeomorphism.

**Riemannian metric learning and frame fields.**   Rather than learning an explicit mapping, some approaches directly parameterize local geometric quantities such as the Jacobian or the metric tensor. JacNet (Lorraine & Hossain, 2024) learns the Jacobian of a function directly, but does not enforce integrability or orthogonality constraints. A separate line of work learns Riemannian metrics on latent spaces via the pullback of the decoder (Arvanitidis et al., 2018; 2020; Gruffaz & Sassen, 2025), providing geometrically meaningful distances and geodesics, but these approaches do not constrain the metric to be conformal or enforce identifiability. Our approach learns a conformal pullback metric $g = \lambda^2 I$ that is highly constrained by construction to combine the geometric flexibility of local metric parameterization with the identifiability guarantees of conformal ICA.

## B    PROOFS

*Proof of Proposition 1.* Expanding the Lie bracket $[\lambda U_i, \lambda U_j]$ component-wise,

$$
\begin{aligned}
[\lambda U_i, \lambda U_j]^k &= \lambda U_i^m \, \partial_m(\lambda U_j^k) - \lambda U_j^m \, \partial_m(\lambda U_i^k) \\
&= \lambda^2 \big(U_i^m \, \partial_m U_j^k - U_j^m \, \partial_m U_i^k\big) + \lambda\big((U_i^\top \nabla \lambda)\, U_j^k - (U_j^\top \nabla \lambda)\, U_i^k\big) \\
&= \lambda^2 [U_i, U_j]^k + \lambda\big((U_i^\top \nabla \lambda)\, U_j^k - (U_j^\top \nabla \lambda)\, U_i^k\big).
\end{aligned}
\tag{8}
$$

Setting this to zero, dividing by $\lambda^2 > 0$, and using $\nabla\lambda/\lambda = \nabla \log \lambda$ gives

$$
[U_i, U_j] = (U_j^\top \nabla \log \lambda)\, U_i - (U_i^\top \nabla \log \lambda)\, U_j.
\tag{9}
$$

Conversely, substituting this relation back recovers $[\lambda U_i, \lambda U_j] = 0$.                $\square$

*Proof of Proposition 2.* We contract the violation tensor with the random vectors and define

$$
\hat{\mathcal{L}}_{\text{int}} = \frac{1}{2} \left\| \sum_{i,j} V_{ij}\, X_i\, Y_j \right\|^2.
\tag{10}
$$

Expanding the squared norm and taking expectations,

$$\mathbb{E}\left[\hat{\mathcal{L}}_{\text{int}}\right] = \frac{1}{2} \sum_{i,j,i',j'} \mathbb{E}[X_i X_{i'}] \, \mathbb{E}[Y_j Y_{j'}] \, \langle V_{ij}, V_{i'j'} \rangle = \frac{1}{2} \sum_{i,j} \|V_{ij}\|^2 = \mathcal{L}_{\text{int}} \tag{11}$$

where the last equality uses the antisymmetry $V_{ji} = -V_{ij}$ and $V_{ii} = 0$. To evaluate the contraction efficiently, form $\tilde{X} = U(\mathbf{x})X$, $\tilde{Y} = U(\mathbf{x})Y$ and define $\alpha = (\nabla_{\mathbf{x}} \log \lambda)^\top \tilde{Y}$, $\gamma = (\nabla_{\mathbf{x}} \log \lambda)^\top \tilde{X}$. Since $X$ and $Y$ are constant vectors, the contraction $\sum_{i,j} V_{ij} X_i Y_j$ reduces to

$$\hat{\mathcal{L}}_{\text{int}} = \frac{1}{2} \left\| [\tilde{X}, \tilde{Y}] - U(\mathbf{x})(\alpha X - \gamma Y) \right\|^2 \tag{12}$$

where $[\tilde{X}, \tilde{Y}] = (\nabla_{\mathbf{x}} \tilde{Y})\tilde{X} - (\nabla_{\mathbf{x}} \tilde{X})\tilde{Y}$ is the Lie bracket of the random vector fields. All terms are computable via Jacobian-vector products, giving $O(n)$ cost per sample. $\qquad\square$

*Proof of Proposition 3.* We first derive the scaled Hessian $\tilde{H}$. Expanding the directional derivative $J_i(J_j(\phi)) = \lambda U_i^\top \nabla_{\mathbf{x}} [\lambda U_j^\top \nabla_{\mathbf{x}} \phi]$ by the product rule and dividing by $\lambda^2 > 0$ gives

$$\tilde{H}_{ij} = U_i^\top \nabla_{\mathbf{x}} (U_j^\top \nabla_{\mathbf{x}} \phi) + (U_i^\top \nabla_{\mathbf{x}} \log \lambda)(U_j^\top \nabla_{\mathbf{x}} \phi) \tag{13}$$

which satisfies $J_i(J_j(\phi)) = \lambda^2 \tilde{H}_{ij}$, so $\tilde{H}_{ij} = 0$ if and only if equation 6 holds. For the masking, since $M_i \in \{0, 1\}$ we have $M_i(1 - M_i) = 0$, so $X_i' Y_i' = X_i M_i Y_i (1 - M_i) = 0$ and the contraction $R = X'^\top \tilde{H} Y' = \sum_{i \neq j} X_i M_i \tilde{H}_{ij} Y_j (1 - M_j)$. Taking the expectation of $R^2$ and using independence of $X, Y$, and $M$,

$$\mathbb{E}[R^2] = \sum_{i \neq j} \mathbb{E}[X_i^2 M_i^2] \, \tilde{H}_{ij}^2 \, \mathbb{E}[Y_j^2 (1 - M_j)^2] = \frac{1}{4} \sum_{i \neq j} \tilde{H}_{ij}^2 \tag{14}$$

where $\mathbb{E}[X_i^2 M_i^2] = \mathbb{E}[X_i^2] \, \mathbb{E}[M_i] = \frac{1}{2}$ and likewise $\mathbb{E}[Y_j^2 (1 - M_j)^2] = \frac{1}{2}$. Therefore $\mathbb{E}[4R^2] = \sum_{i \neq j} \tilde{H}_{ij}^2 = \mathcal{L}_{\text{ind}}$. The contraction $R$ can be written as $X'^\top \tilde{H} Y' = (UX')^\top \nabla_{\mathbf{x}} [(UY')^\top \nabla_{\mathbf{x}} \phi] + [(UX')^\top \nabla_{\mathbf{x}} \log \lambda] [(UY')^\top \nabla_{\mathbf{x}} \phi]$, which requires only two Jacobian-vector products, giving $O(n)$ cost per sample. $\qquad\square$

