# OpenReview forum: "Conformal Coordinate Frames for Disentanglement"
_ICLR.cc/2026/Workshop/GRaM — ICLR 2026 Workshop GRaM Poster_

### Official Review · Reviewer_3SB6 · 2026-02-16
**Nicely written, limited technical novelty and applicability**

**Rating:** 5
**Confidence:** 2

**Review:**

Summary. The paper proposes learning local conformal frame fields J(x)=lambda(x)U(x)as a scalable relaxation of conformal ICA/IMA, enforcing distribution matching (via score matching), integrability (Lie brackets) and independence (off-diagonal latent Hessian) using stochastic losses that rely only on Jacobian–vector products. Experiments on rotated Neal’s funnel (4–64D) show high alignment with the true funnel axis and outperform a globally conformal Möbius baseline.

Major issues: (1) The empirical evidence is too narrow: essentially one synthetic family (Neal’s funnel) and one metric (“axis alignment”), so it is hard to judge disentanglement beyond recovering a single dominant direction. (2) The current method is not unsupervised in the usual sense: the paper notes a key limitation that the independence loss requires access to the true log-density (or equivalent second-order structure), which strongly restricts applicability and weakens the “practical foundation” framing. (3) Technical novelty is incremental: the core idea is a constrained Jacobian/frame-field parameterization with stochastic contractions for efficiency; this is interesting, but the paper does not convincingly establish how it compares empirically to nearby approaches (e.g., IMA/PCF-style or Jacobian-learning baselines) on the same task.

Minimal suggestions: (1) Add at least one additional distribution/task beyond Neal’s funnel (e.g., multi-factor synthetic with >1 meaningful direction) and report a disentanglement-style metric that checks multiple components, not only top-axis alignment. (2) Report a concrete applicability demo: show how to use the learned frame for manipulation (local traversals) and quantify the effect; additionally, verify integrability by integrating the frame locally and measuring consistency/commutation. (3) Strengthen baselines: compare to a simple Jacobian/metric-learning baseline (and/or an IMA/PCF-like method) under matched settings.

Overall: The geometric framing is neat and the stochastic O(n) losses are a good engineering contribution, but the current submission feels closer to a proof-of-concept on a single synthetic benchmark with limited demonstration of disentanglement or practical use, and with a key assumption (access to log-density) that sharply limits scope.

**Pmlr Suitability:**

NA

---

### Official Review · Reviewer_jVW4 · 2026-02-16
**Useful Extension of IMA with Local Conformal Frames**

**Rating:** 7
**Confidence:** 2

**Review:**

**Summary:**

This paper proposes a method for unsupervised disentanglement by learning a local conformal frame field rather than a global conformal map. The authors extend the IMA principle to a local setting, avoiding the rigidity of Liouville’s Theorem in dimensions $n \ge 3$.

The method enforces three constraints: distribution matching via score matching, integrability (vanishing Lie brackets), and statistical independence of the latent Hessian. What I find interesting is their proposal of using stochastic estimators for the integrability and independence losses that scale linearly with dimension ($O(n)$), making the method tractable. Experiments on Neal’s Funnel demonstrate that the local approach recovers the ground truth structure where global conformal maps fail.


**Strengths:**

- **Practical & Scalable Extension of IMA:** The move from global to local conformal constraints is a smart way to bypass theoretical bottlenecks (Liouville's Theorem) while retaining the useful inductive bias of orthogonality. The derivation of stochastic loss terms that require only Jacobian-vector products is a significant contribution, enabling scalability to higher dimensions ($D=64$).

- **Clear Problem Motivation:**
The paper clearly articulates why standard Conformal ICA fails in practice and how local frames offer a flexible alternative.

- **Promising Proof-of-Concept:**
While the experiments are limited to toy data (Neal's Funnel), this is appropriate for a Tiny Paper track. The results clearly demonstrate the failure mode of the global baseline and the success of the local method.

**Weaknesses & Questions**
I am leaning towards acceptance, but I have a few questions and suggestions that would strengthen the final manuscript or future versions of this work.

*1. End-to-End Pipeline & Parametrization:*

While the theoretical derivation of the losses is clear, the implementation details of the neural parametrization could be more explicit. Specifically, the "end-to-end" pipeline—from input $x$, through the neural network parametrizing $U(x)$ (via QR decomposition) and $\lambda(x)$, to the computation of the stochastic losses—is vital for reproducibility.

*Suggestion:* Adding a schematic diagram or pseudocode (even in the appendix) would be very helpful. It should explicitly show how the QR decomposition is integrated into the forward pass to ensure $U(x) \in O(n)$ and how the gradient flows back from the stochastic estimators.

*2. Baseline Behaviors in Figure 2:*

The behavior of the baselines in Figure 2 raises some interesting questions:

- Random Baseline: The alignment for the random baseline remains surprisingly high even at $D=64$ (~0.25). Is this purely an artifact of using the "maximum absolute inner product" metric? Heuristically, with 64 random vectors, the probability that one of them aligns somewhat well with the ground truth is non-negligible. A brief comment on whether this matches the expected value from random matrix theory would clarify if this baseline is strong or weak.

- Global Baseline Non-Monotonicity: The global conformal baseline shows a non-monotonic trend: it starts high at $D=4$, drops at $D=8$, and then appears to rise again at $D=64$. I would have expected a monotonic increase (as the space grows larger, vectors get more "gaussian") or a consistent failure. Do the authors have a heuristic explanation for this "dip and recovery"?

*3. Dynamics of the Integrability Loss*

The ablation study (Figure 3) suggests that the integrability loss $L_{int}$ has a negligible effect on alignment. While I understand the intuition—one can find the "principal direction" (via $L_{ind}$) without successfully "stitching" the local frames into a valid global coordinate system—it would be illuminating to see the training dynamics.

*Suggestion:* A plot showing the values of the three loss components ($L_{score}$, $L_{int}$, $L_{ind}$) over the course of training would help confirm this hypothesis. Does $L_{ind}$ drop quickly while $L_{int}$ lags behind?

**Minor Comments**
- Alignment Metric: The "maximum absolute inner product" is a generous metric. While might be standard for identifying a single factor, have the authors considered reporting the mean absolute inner product? This would reveal if the model is recovering only the funnel axis while outputting noise in the other $n-1$ dimensions, or if the entire frame is meaningful.
- Formatting: In Figures 2 and 3, the legend box obscures the data bars. Moving the legend outside the plot area would improve readability.

**Pmlr Suitability:**

NA

---

### Official Review · Reviewer_WtLA · 2026-02-24
**Sound idea, meaningful modeling, with open challenges**

**Rating:** 6
**Confidence:** 3

**Review:**

The paper focuses on the problem of disentanglement of latent representations, in particular from the perspective of independent mechanisms. The proposed method aims to learn, directly in the data space, the Jacobian of a conformal map from a latent space. The key idea is that this map, and consequently its Jacobian, only needs to be locally conformal, and that this local property is sufficient for recovering the principal directions of data variation. The Jacobian to be learned is designed to satisfy local conformality, and the learning process to ensure that it corresponds to a coordinate vector field, that the latent codes although not explicitly learned are independent, and that the model fits the given density. The efficiency of the method is demonstrated on a known distribution.

The paper aligns well with the topic of the workshop, as it approaches the disentanglement problem from a geometric perspective. The idea appears to be original and extends previous works that establish conformal maps as a tool for disentanglement. I find particularly interesting the condition ensuring that the learned Jacobian corresponds to a coordinate vector field.

The technical part of the paper appears sound, and is well appreciated that theoretical support is included, although I have not checked the proofs in detail. Also, in the experiments the behavior of the model across different dimensions and regularization levels is provided.

The paper is generally well written, but some clarifications would be helpful (see questions below).

1. It seems that the Jacobian is learned directly in the data space, and that the generative map itself is not required. In particular, in Eq. 1 the term $\log p_z$ is not utilized under the assumption of a uniform distribution. However, in Eq. 6 the latent log-density $\phi$ is considered. I would appreciate some clarification on this point.

2. Regarding the latent derivative operator, I would expect it to be equal to $M^{-1} J^T \nabla_x$, where $M$ is the associated metric tensor. Please clarify this as well.

The downsides and challenges are, as the authors mention in the limitations, the dependence on the true probability density function, as well as the fact that local conformality is only conjectured to be sufficient for disentanglement, even though this seems reasonable. Also, the experiments, even if nicely conducted, are somewhat limited.

**Pmlr Suitability:**

NA

---

### Meta-Review · Area_Chair_Jkac · 2026-02-25

**Decision:**

Accept

**Metareview:**

This is a neat tiny paper proposing a conformal frame field approach for disentanglement. It bypasses rigid global constraints and is mathematically sound. The empirical part is a bit limited (just Neal's Funnel) but for a tiny paper, this is perfectly fine and a solid contribution to the workshop theme. Happy to accept.

**Relevance To Proceedings:**

Tiny paper — does not apply

**Relevance To Workshop:**

Yes — suitable for GRaM

---

### Decision · Program_Chairs · 2026-03-02

Accept (Poster)